# Simultaneous Isolation and Purification of Transferrin and Immunoglobulin G from Human Serum—A New Biotech Solution

**DOI:** 10.3390/molecules30050993

**Published:** 2025-02-21

**Authors:** Danilo Četić, Goran Miljuš, Zorana Dobrijević, Nikola Gligorijević, Aleksandra Vilotić, Olgica Nedić, Ana Penezić

**Affiliations:** Institute for the Application of Nuclear Energy INEP, University of Belgrade, Banatska 31 b, 11000 Belgrade, Serbia; danilo.cetic@inep.co.rs (D.Č.); goranm@inep.co (G.M.); zorana.dobrijevic@inep.co.rs (Z.D.); nikola.gligorijevic@ihtm.bg.ac.rs (N.G.); aleksandrav@inep.co.rs (A.V.); olgica@inep.co.rs (O.N.)

**Keywords:** biotherapeutics, isolation and purification, ion-exchange, IgG, transferrin

## Abstract

A fast and simple biotech method is presented for the simultaneous isolation and purification of transferrin (Tf) and immunoglobulin G (IgG) from the same pool-sample of human serum, yielding >98% pure proteins. Serum sample preparation was achieved by precipitation with ethacridine lactate (rivanol). Protein purification was performed with AKTA Avant 150 FPLC, using a Resource Q column. Three different buffers at pH 6.2 (MES, phosphate, and Bis-Tris) were tested. Isolated and purified proteins retained their native 3D structure, as shown by spectrofluorimetric measurements. Tf functionality was preserved, as confirmed by the retention of both the iron binding capacity and its ability to interact with the transferrin receptor (immunofluorescent staining), as well as the immunogenicity of IgG, as shown by Western blot analysis with immunodetection. The formation of IgG aggregates was avoided. This biotech method is a rapid, simple, and time-saving alternative to other methods for the isolation of extremely pure IgG and Tf, while it is also the only method so far described for their simultaneous isolation.

## 1. Introduction

Human transferrin (Tf) is a 76 kDa, bilobal, iron-binding glycoprotein mainly synthesized by hepatocytes [1,2]. It is present in human serum (approximately 2.5 g/L in healthy adult individuals) but also in interstitial and cerebrospinal fluids as well as lymph [2,3,4]. In physiological conditions, the main role of Tf is to transport and deliver ferric ions through circulation to cells via interaction with its receptor on the cell surface [2]. Human serum transferrin applications include a source of iron in cell culture media [5], targeted drug delivery [6], a therapeutic in the treatment of atransferrinemia, iron-dependent anemia, oxidative stress and reperfusion injury [7], and cancer [8].

Human immunoglobulin G (IgG) is a 150 kDa glycoprotein belonging to the immunoglobulin family [9]. It is one of the most abundant serum proteins with a physiological concentration range of 10–25 g/L in healthy individuals [10]. Physiological roles of IgG antibodies include the neutralization of different toxins, viruses, and bacteria, opsonization, and the activation of the complement system [11]. IgG antibodies are biopharma industry highflyers, especially with taking the COVID-19 epidemic into consideration. Polyclonal IgG isolated from the human serum is widely used as a replacement treatment and for infection prevention in immunodeficient patients and as a therapeutic in the treatment of autoimmune and acute inflammatory diseases [12,13,14].

Methods for the separate purification of both IgG [15,16,17] and Tf [18,19,20,21] have been described so far, utilizing affinity or ion-exchange chromatography with different sample preparation steps, such as ethanol [19] and ammonium sulphate precipitation [17]. However, the simultaneous purification of these two proteins from the same, single sample has not been described so far. A comparison of methods using human serum, plasma, or Cohn fractions of human plasma available in literature to our method is provided in Table 1.

Given that 80% of the production cost goes to the downstream processing, i.e., purification and polishing, of any protein biopharmaceutical, a lot of research has been conducted to reduce these costs [28]. On the other hand, the native structure and physiological functionality of isolated transferrin and immunoglobulin G is seldom checked. As far as the state-of-the-art goes, for the isolation and purification of transferrin and IgG from human serum in a simultaneous fashion and from the same sample, the literature survey gave no information (Table 1). The largest number of papers focuses on the recombinant production of a single protein and the downstream processing from cell media or the fermentation process.

Here, we describe a fast and simple FPLC method for the simultaneous isolation and purification of both proteins from a single sample. The method consists of a single precipitation step, followed by a single chromatographic step, yielding Tf and IgG of high purity. Figure 1 shows the entire purification process in flowchart form. This method not only allows for both proteins to be isolated from the same sample but also to retain their native 3D structure and physiological properties. In turn, the proposed method is enabling the production of two theranostic-grade proteins applicable in research, analytical procedures, diagnosis, and therapy from the same sample simultaneously.

## 2. Results and Discussion

### 2.1. Protein Precipitation

Rivanol precipitation was used primarily to remove human serum albumin (HSA) from the serum pool, as well as other α– and β–globulins [17]. Figure 2 shows electrophoretic protein profiles in the initial serum pool and supernate after the precipitation and removal of rivanol by activated charcoal. Molecular weight markers were used for preliminary protein identification, having in mind the electrophoretic profile of human serum. The protein precipitation step allowed for a significant reduction in the HSA content and the partial purification of proteins of interest from the initial serum pool.

Buffers at pH 6.2 were chosen because of the pI values of Tf and IgG. The pI values of holo-Tf and apo-Tf are 5.2 and 5.9, respectively [20]; whereas, IgG has a pI range between 7.0 and 9.95 (isotypes with pI lower than 7.0 are far less abundant) [29]. At the given pH of 6.2, Tf is expected to bind to the resin; whereas, IgG is expected not to interact and, thus, be eluted in the wash step. This allows for the immediate isolation and purification of IgG. Following the elution step, pure Tf is obtained. Respective chromatograms are presented in Figure 3. Chromatographic purification using each buffer was carried out in an intraday triplicate, as well as an interday triplicate (i.e., three different chromatographies were performed in one day, as well as one per day in three consecutive days). Inter-/intraday reproducibility was assessed by the peak area and retention volume data, presented in the form of the mean followed by the % of coefficient of variability in Table 2. A very low % of coefficient of variability confirms the reproducibility of the chromatographic purification step of this method both in interday and intraday replicates.

This method has already been successfully applied for the isolation and purification of Tf and IgG from different serum pools of healthy individuals (Figure 3), as well as from the pools of patients with different pathologies, i.e., diabetes mellitus and kidney disease, with consistent results in terms of purity and yield. In addition, these experiments were performed by using the same IEX column at a higher capacity, thus introducing a version of scaling-up for this method. The representative chromatogram of Tf and IgG purification from diabetic serum pool is shown in Figure 4. Of course, due to the scaling-up process, the elution profile was slightly modified, because we excluded the 75% B step as unnecessary and also increased the duration of each respective step (see Figure 4) in order to achieve better separation.

The chosen pH of 6.2 enables the use of different buffers, providing added flexibility to this method.

The first peak in all three chromatograms presented in Figure 3 corresponds to IgG, and the last peak is pure Tf, as shown by silver staining and Western blotting with immunodetection (Figure 5, Figure 6 and Appendix A). The difference in Tf elution profile when using PB is most likely due to the higher ionic strength of PB in comparison to the other two buffers. The entire chromatographic process (system and column equilibration, sample application, column washing, and elution) takes under an hour and a half to complete.

### 2.2. Protein Yield, Purity, Purification Factor, and Immunogenicity Testing

The purity of isolated IgG and Tf was determined by the densitometric evaluation of the silver-stained SDS PAGE gels (Figure 5). Densitometric analysis was carried out in triplicate. As can be seen, highly pure proteins were obtained using all of the tested buffers (Table 3). The yields of IgG and Tf were calculated after the determination of their concentration in the initial serum pool and in chromatographic fractions. The highest yield of IgG was obtained when PB was used, while the application of BT buffer resulted in the highest yield of Tf (Table 3). All results are presented as mean ± standard deviation (where possible). Purification factor, as a measure of the efficiency of this method, was calculated by dividing the protein purity in the given step with the protein purity in the initial sample.

Relatively low yields of IgG and Tf can be partially attributed to the loss of the target proteins during precipitation with rivanol. This is because rivanol not only precipitates HSA specifically from serum but other proteins as well [30]. However, the loss in the yield is mostly the due to protein adsorption on the activated charcoal, which has been described in literature, and seems to be the case in this method, as we used it to remove the excess of rivanol [31]. Moreover, the wide pI range of IgG is an important factor impacting IgG yield, as some isoforms bind to the column and are eluted in different parts of the elution phase (i.e., the unmarked peaks on the chromatograms of Figure 2). Currently, we are testing alternative methods and solutions for excess rivanol removal, which would provide a higher yield. Also, we already have undergoing experiments for the testing of other solutions for the protein precipitation step in order to avoid the usage of rivanol. Additionally, we are undertaking promising scale-up experiments involving an eight times larger volume column of similar properties (IEX), in order to isolate and purify IgG and Tf in larger quantities, while retaining extremely high purity and the native structure of the isolates.

To show the protein identity (IgG and Tf) and immunogenicity of IgG, nitrocellulose membranes were incubated with anti-Tf and anti-IgG antibodies (FITC labeled), followed by the incubation with secondary antibodies (for Tf detection) and ECL for visualization. The results of this analysis (Figure 6A—IgG and Figure 6B—transferrin) confirmed the immunogenicity originating from fully preserved epitopes on the isolated IgG (recognized by the sheep anti human IgG antibody), as well as the identity of both isolated proteins. As shown in Figure 6, the utilization of this new biotech method enables the isolation of Tf and IgG from the starting serum pool.

### 2.3. IgG Aggregation Analysis—Aggregation Index (AI)

The presence of the aggregates or high-order physiologically irrelevant oligomers represents a serious problem in downstream the processing of pharmaceutically relevant proteins. If present, aggregates must be removed, usually by size-exclusion chromatography, thus adding another step in the purification protocol. This makes the procedure more time consuming and also leads to even lower yields. Upon isolation and purification, quality control should be performed, including the testing for homogeneity and the native state status of the isolated IgG. UV/Vis spectroscopy is commonly used for the assessment and quantification of aggregates in IgG isolates [28], since the absorbance of the sample at wavelengths greater than 320 (namely 340 nm) can be attributed to particles with a high hydrodynamic radius, i.e., aggregates, and not to native protein particles. The formula for the calculation of the aggregate percentage in the sample is presented in the Materials and Methods section. Following the isolation and purification procedure described in this paper, the IgG fraction was tested for the presence of aggregates. The AI index value obtained for each buffer system was 3.43%, 3.65%, and 3.92% for BT, MES, and PB, respectively, showing that the final IgG product can be considered to be aggregate-free. Having in mind that the aggregation process depends on the temperature, the pH of the buffer system, ionic strength, protein concentration, mechanical stress, and surface interactions [32], the parameters used in this isolation and purification protocol are well chosen, and the formation of IgG aggregates is avoided.

### 2.4. Testing of Tf Functionality

The iron-binding capacity of isolated and purified Tf was analyzed by determining the iron concentration in the purified Tf (pTf) sample by the commercially available ferrozine method kit, used for the diagnostic in vitro quantification of iron in human serum samples. The ferrozine method is based on the application of the monosodium salt hydrate of 3-(2-pyri-dyl)-5,6-diphenyl-1,2,4-triazine-p,p ‘-disulfonic acid reagent, FerroZine, which reacts with iron to give a stable magenta complex. Iron saturation was calculated as the iron ion concentration (µmol/L):transferrin concentration (µmol/L) ratio. The calculated iron saturation of pTf was 29.19%, which corresponds to the physiological value. This result confirmed that this method for the isolation and purification of Tf from human serum does not disturb the binding site for iron ions nor cause iron leakage from the Tf molecule. Having in mind that the iron saturation and iron binding capacity of Tf are extremely important in diagnostics, as well as in the therapy of iron deficiency/overload-related diseases, this method for purification represents an ideal solution for Tf analysis/production precisely because of the full preservation of Tf iron-transporting capacities.

Another important functional feature of the Tf molecule is its interaction with the transferrin receptor as an important step in the iron traffic throughout the organism. The cellular uptake of pTf, which is mediated by TfR1 receptor, was investigated on immortalized human extravillous trophoblast HTR-8/SVneo cells (HTR) as a model system. TfR1 expression was previously shown in HTR cells [33], and it was further confirmed in our study (Figure 7B,E). Control cells, maintained in SFCM only (SFCM controls), exhibited weak cytoplasmic Tf staining, likely originating from the residues of yet unreleased Tf (Figure 7A). However, strong Tf staining localized in larger perinuclear and punctuate structures distributed across the cytoplasm was detected after cell incubation with our pTf sample (Figure 7D). Tf staining was colocalized with TfR1 fluorescence (Figure 7F), implying that pTf kept its functionality (i.e., pTf has maintained its ability to bind to its cognate receptor followed by receptor-mediated internalization). Results from this experiment showed that the isolation and purification procedure did not interfere nor disrupt the interaction between Tf with the TfR1 and, thus, proved that isolated Tf kept its functional integrity. Furthermore, this also proves the ability of isolated Tf to deliver iron in vitro, as Tf/TfR1 interaction is not possible if there is no iron bound to Tf.

### 2.5. Analysis of Protein 3D Structure by Recording Fluorescent Emission Spectra

Fluorescent emission spectra of the intrinsic fluorescence originating from Trp residues in pIgG and pTf (excitation at 295 nm) were taken in order to evaluate if there are any differences in the native conformation between the isolated proteins and their commercially available counterparts. The intensity and position of the peak at the maximum wavelength upon excitation at 295 nm during fluorescent emission are dependent on the surroundings of the fluorophore, in this case, Trp residues. The comparison of the fluorescent emission spectra of apo-pTf and pIgG and their commercially available counterparts (Figure 8A,B) confirmed that the 3D (native) structure of both proteins remained intact—purified proteins have identical fluorescent emission spectra to their commercial counterparts. The spectral shape is consistent between the isolate and the commercial counterpart, maintaining an identical emission maximum wavelength in the case of Tf (324 nm). The peak shift in the case of IgG (327 nm for purified IgG vs. 325 nm for standard IgG) can be explained by the fact that both IgG samples are polyclonal in nature, so different ratios of different IgG forms (subclasses ratios and glycoforms) are present in both samples. These results show that the proposed method and the applied conditions (temperature, buffer choice, ionic strength, and pH) have no effect on the native structure of isolated proteins, i.e., both isolates retain their native conformation. Also, given that the spectra obtained for both purified proteins after approximately 6 months of storage at −4 °C correspond to the spectra of purified proteins closely after purification, both proteins purified using this method are deemed stable after longer periods of storage.

## 3. Materials and Methods

### 3.1. Serum Samples

Serum samples were obtained from healthy adult individuals (35 individuals, 17 male, 18 female; age range—20 to 40 years old) at the Institute for the Application of Nuclear Energy (INEP). The study conformed to the Helsinki Declaration of 1975 (as revised in 2008). Oral informed consent was sought and obtained from all the participants included in this study. The serum pool was created by using 600 µL of each serum sample, reaching the final volume of 21 mL, which was sufficient for all the experiments and replicates. The created serum pool, not individual samples, was used as a starting material for all of the experiments.

### 3.2. Chemicals and Reagents

All chemicals were purchased from Sigma-Aldrich (Stanheim, Germany) unless stated otherwise. MiliQ water was used in all experiments.

### 3.3. Protein Precipitation

The serum pool was treated with ethacridine lactate (rivanol) to precipitate proteins (mainly HSA). A 1.2% rivanol solution adjusted to pH 6.2 was slowly (drop by drop) added to the serum pool in a 1:1 ratio (vol:vol) with constant but gentle stirring. The precipitate was removed by centrifugation at 4500 rpm at 7 °C for 15 min. Following centrifugation, the supernate was treated with activated charcoal in order to remove rivanol from the solution. Finally, the pH of the sample was again adjusted to 6.2.

### 3.4. Isolation and Purification of Tf and IgG by Ion-Exchange Chromatography

Chromatography was performed using AKTA Avant 150 chromatography system (GE Healthcare, Chicago, IL, USA), operated by Unicorn 7.6 software. Chromatographic purification was carried out on a Resource Q ion-exchange column (GE Healthcare, Chicago, IL, USA; V0 = 6 mL). Three different buffers, all 20 mM and at pH 6.2—MES, Bis-Tris (BT), and sodium phosphate (PB)—were tested. Other buffers were also considered, but due to availability and the required pH value, only these 3 were tested. The buffer system used for the elution of the proteins consisted of buffer A and buffer B (buffer B being the same as buffer A, with the addition of 175 mM NaCl). Column equilibration was carried out according to the manufacturer’s instructions for Resource Q column. The sample (1 mL) was applied to the column (flow rate of 0.75 mL/min), followed by column washing with buffer A (flow rate of 1.5 mL/min; 3 column volumes). Elution was performed in a stepwise manner, using the gradient concentration of NaCl at discrete elution steps (25, 50, 75, and 100% of buffer B; flow rate of 1.5 mL/min; 2 column volumes per step). Elution was monitored by recording the absorbance at 280 nm, and peak fractions (2 mL) were collected. The reproducibility of the method was tested, for each buffer, in an intraday triplicate, as well as in three consecutive days (interday triplicate).

### 3.5. Analysis of Isolated and Purified Proteins by SDS PAGE and by Western Blot with Immunodetection

The purity of eluted fractions was analyzed by SDS PAGE (under reducing conditions; 10% PAA gels loaded with 15 μL of sample per well), followed by silver staining. Silver staining was performed using a silver stain kit (Bio-Rad Laboratories Inc., Hercules, CA, USA), according to the manufacturer’s instructions. The Blue Wide Range Protein Ladder was used as molecular weight marker (Cleaver Scientific Ltd., Rugby, UK; range: 10–245 kDa). SDS PAGE was performed at 200 V for 1 h according to Laemmli. Following SDS PAGE, proteins were electrotransferred to nitrocellulose membrane (Western blot; semi-dry conditions for 1 h at 20 V; Trans-Blot SD Semi-Dry Transfer cell, Bio-Rad Laboratories Inc., Hercules, CA, USA) for immunodetection. The efficacy of electrotransfer was assessed by Ponceau S staining. The immunodetection of isolated and purified protein species was performed by using polyclonal sheep anti-human Tf antibodies (INEP, Belgrade, Serbia, 1:5000) and polyclonal sheep anti-human IgG antibodies (INEP, Belgrade, Serbia, 1:2000) and appropriate secondary antibodies (Vector Laboratories Inc., Newark, CA, USA, 1:10,000). The visualization of protein bands was achieved using ECL Western Blotting Substrate (Thermo Fisher Scientific, Waltham, MA, USA). The intensities of the protein bands were determined using the ChemiDoc MP Imaging System (Bio-Rad Laboratories Inc., Hercules, CA, USA). The purity and yield of purified and isolated proteins was determined by densitometry using Image Lab Software version 6.1.0 (Bio-Rad Laboratories Inc., Hercules, CA, USA). The same concentration of all samples was used for electrophoretic/blotting analysis, so there was no difference in the protein concentration between lanes.

### 3.6. Determination of Tf and IgG Concentrations

Tf concentration was determined by commercial immunoturbidimetric kit (Bioanalytica, Belgrade, Serbia) on Biossays 240 Plus biochemical analyzer (Snibe Diagnostic, Shenzhen, China). IgG concentration was determined by commercial radioimmunodiffusion test (INEP, Belgrade, Serbia).

### 3.7. IgG Aggregation Analysis

IgG aggregation was assessed by the aggregation index (AI) value obtained by recording the absorbance of the sample at 265 and 340 nm (UV/VIS). AI is predominately affected by the particle size, meaning that, the larger the molecule (particles with hydrodynamic radius greater than 200 nm can be considered high-order oligomers or aggregates), the greater the light scattering intensity and, consequently, the greater value of the AI. Solutions with AI values below 10 are considered as solutions with insignificant amounts of soluble aggregates [34]. AI was calculated as:(1)$$AI=100\times \frac{A_{340}}{A_{265}-A_{340}}$$

### 3.8. Assessment of Tf Iron-Binding Capacity and the Interaction with Transferrin Receptor 1

To confirm if the isolated transferrin retained its functionality, we tested:Iron-binding capacity, by measuring the iron content in the isolated transferrin sample. This was achieved by applying the commercially available ferrozine reagent kit supplied with standard solution/calibrator (Biosystems, Barcelona, Spain, code 12509) on a BioSystems A25 Analyzer (Biosystems, Barcelona, Spain).The interaction with transferrin receptor 1 (TfR1) expressed on the surface of human extravillous trophoblast HTR-8/Vneo cells via immunofluorescence staining in vitro. The cells were seeded in 24-well plate on glass coverslips (1 × 10^5^ cells/well) in culture medium consisting of RPMI 1640 medium (Gibco, Paisley, UK), supplemented with 10% heat inactivated fetal calf serum (*v*/*v*) (FCS, Sigma Aldrich, St. Louis, MO, USA) and 1% antibiotic/antimycotic solution (Capricorn Scientific, Ebsdorfergrund, Germany) overnight at 37 °C. The following day, after rinsing with phosphate buffer saline (PBS), cells were incubated for 2 h in serum-free culture medium (SFCM) in order to remove Tf of FCS origin from the cells. Subsequently, the medium was discarded, and cells were treated with Tf purified using BT buffer (pTf, 0.1 mg/mL) in serum-free culture medium for 2 h. Control cells were kept in SFCM only. After incubation, cells were first rinsed with PBS, fixed in 4% paraformaldehyde for 15 min at room temperature (RT), and permeabilized with 0.1% of Triton ×100 in PBS for 10 min at RT. Following this, non-specific antibody binding was blocked with 1% bovine serum albumin in 0.05% Tween in PBS for 1 h at RT. The incubation of cells with primary anti-human Tf antibodies (1:50, INEP, Serbia) and anti-human TfR1 antibodies (1:100, Santa Cruz Biotechnology Inc., Dallas, TX, USA) took place in a humidified chamber overnight at 4 °C. The visualization of primary antibody binding was carried out using anti-goat Alexa 488 and anti-mouse Alexa 555 antibodies (both at 1:1000, Invitrogen, Waltham, MA, USA), respectively, which were incubated with the samples for 1 h at RT. Non-specific binding controls were prepared without primary antibodies. Nuclei were counterstained with Vectashield Mounting Medium with DAPI (Vector Laboratories, Newark, CA, USA). Images of the samples were taken using 40× objective on a Carl Zeiss Axio Imager.A1 microscope with AxioCam MRm camera (Carl Zeiss, Oberkochen, Germany).

### 3.9. Spectrofluorimetric 3D Structure Analysis of Tf and IgG

A protein 3D structure was confirmed by recording fluorescence emission spectra and comparison of both IgG and apo-Tf with their commercially available purified protein counterparts. The conversion of pTf to apo-pTf, i.e., the removal of Fe^3+^ ions, was accomplished by the addition of a 5 mM EDTA solution at pH 5.0 in a 1:1 ratio, after which the solution was incubated at 4 °C for 48 h. EDTA and iron were then removed from the solution by using 10 kDa cutoff centrifugation filters (Amicon^®^ Ultra-15 Centrifugal Filter Unit, Merck, Darmstadt, Germany), and the pH was readjusted to 6.2. The same purified protein samples were kept at −4 °C for approximately 6 months and re-evaluated in the same way. The emission fluorescence spectra of both proteins (isolated and commercial) were recorded on a RF-6000 Series Spectrofluorometer with LabSolutions RF Software version 1.15 (Shimadzu, Kyoto, Japan) at 0.6 µM protein concentration in the 300 to 500 nm range, following excitation at 290 nm. A 1 cm path length quartz cell was used, while slit widths were 4 nm. The resulting fluorescence emission spectra represent the average of three consecutive recordings.

Calculations:(2)$$Purity\left(\%\right)=\frac{Amount\;of\;purified\;protein\;(mg)}{Amount\;of\;total\;protein\;(mg)}\times 100$$(3)$$Yield\left(\%\right)=\frac{Amount\;of\;purified\;protein\;\left(mg\right)}{Amount\;of\;protein\;in\;initial\;serum\;pool\;\left(mg\right)}\times 100$$(4)$$Purification\;factor=\frac{Purity\;of\;purified\;protein\;(\%)}{Purity\;of\;protein\;in\;initial\;serum\;pool\;(\%)}$$

## 4. Conclusions

The method described in this paper has been developed as a rapid, simple, timesaving, and cost-effective alternative to other methods used only for the isolation of either Tf or IgG but not both simultaneously. Moreover, this is the only method so far described for the simultaneous isolation and purification of these two proteins. The entire process from sample preparation to the end of the elution lasts approximately 80 min, which makes this method a fast solution for the isolation and purification of extremely pure IgG and Tf. This is in stark contrast to other methods, which can be extremely time- and labor-intensive, such as protocols that use Cohn fractionation for serum sample preparation [19,20], while our method also provides the added benefit of the simultaneous isolation of these two proteins. All of the methods shown in Table 1 require at least 2 h to complete, while some (e.g., those using Cohn fractionation) require far more than that. Additionally, only one of these methods provides protein purity at the level our method provides, while no method in the literature (including those shown in Table 1) provides the simultaneous purification of IgG and Tf at >99% purity.

Our method can easily be adjusted for bioanalytical purposes, which can be extremely useful for the differential diagnostic procedures used for the analysis and monitoring of aberrant glycosylation in certain diseases. In addition to protein purity and the 3D structural integrity of both proteins, Tf obtained by this method was assessed in terms of functionality, proving that the protein retains both of its main functions (iron binding and binding to TfR1); whereas, protein functionality is rarely described in other established methods. Likewise, purified IgG retained its immunogenicity and was essentially aggregate-free, thus presenting a fully functional product with no further need of additional processing and polishing steps.

## Figures and Tables

**Figure 1 molecules-30-00993-f001:**
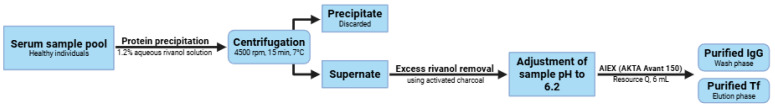
Flowchart showing the entire purification process from serum sample pool to purified IgG and Tf.

**Figure 2 molecules-30-00993-f002:**
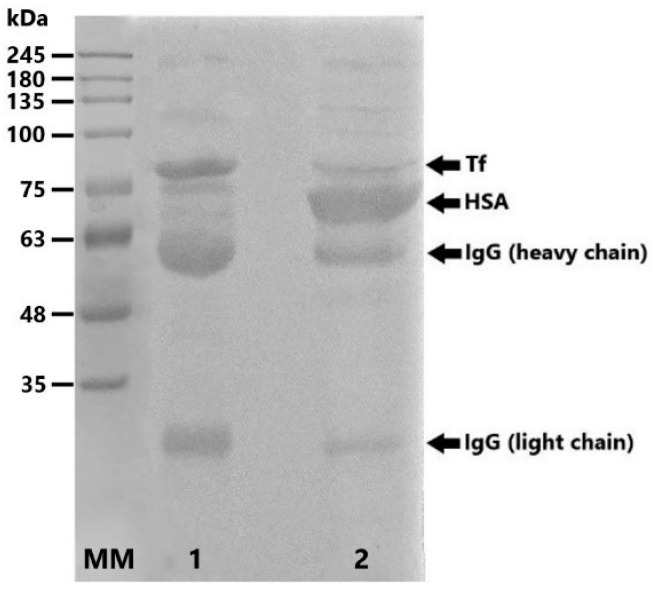
Ponceau S-stained nitrocellulose membrane showing 10% SDS PAGE (reducing conditions) electrotransfer of starting serum pool (lane 2) and sample after rivanol treatment (lane 1). The starting pool was additionally diluted for analysis to achieve similar band signal.

**Figure 3 molecules-30-00993-f003:**
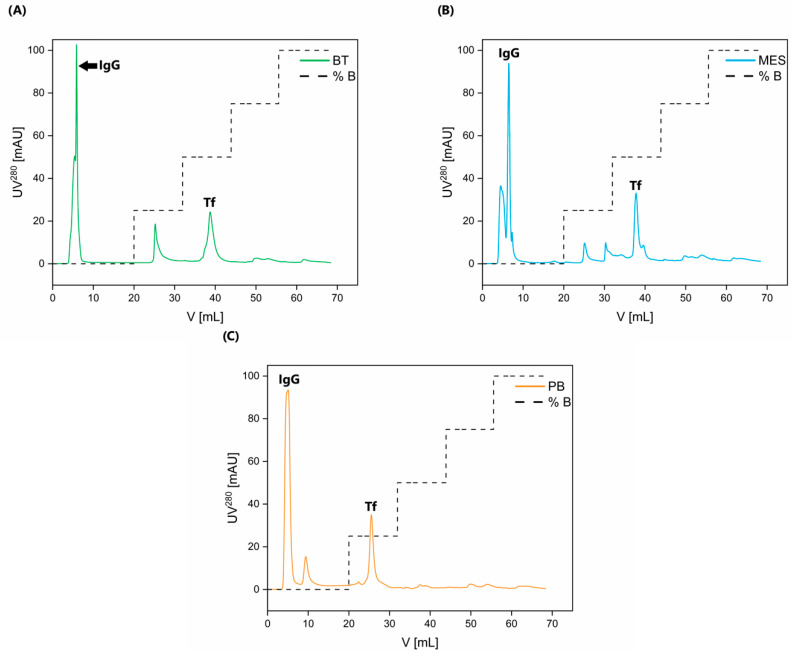
Respective chromatograms of eluted proteins using (**A**) BisTris buffer (BT), (**B**) 2-(N-morpholino)ethanesulfonic acid buffer (MES), and (**C**) sodium phosphate buffer (PB). Elution steps indicate 25, 50, 75, and 100% buffer B or from 0 mM to 175 mM sodium chloride. IgG and Tf peaks are marked (IgG elutes in the wash phase, hence it does not bind to the column, while Tf elutes in the 50% B step for BisTris and MES or 25% for sodium phosphate buffer), while the unmarked peaks represent impurities (mainly HSA, as well as some IgG isoforms with inadequate pI values for wash phase elution).

**Figure 4 molecules-30-00993-f004:**
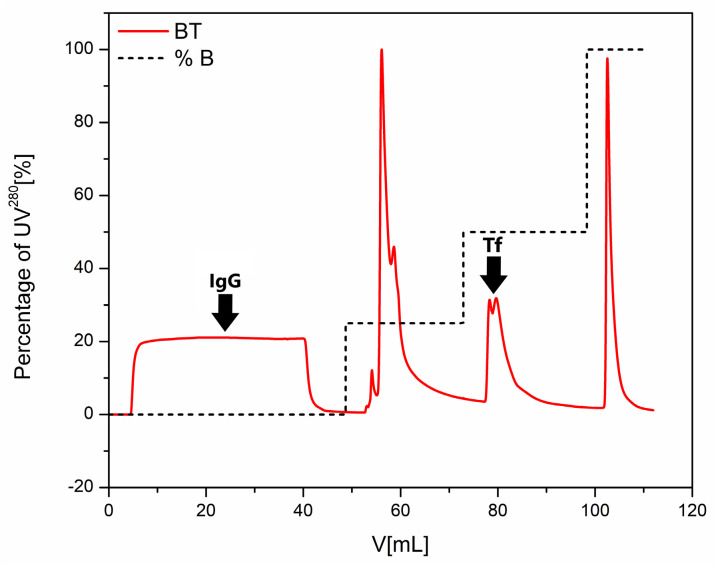
Chromatogram of the purification process of IgG and Tf from serum pool of diabetic patients using BisTris buffer. Different peak intensities compared to healthy IgG and Tf purification are due to differences in the original properties between the healthy and the diabetic samples.

**Figure 5 molecules-30-00993-f005:**
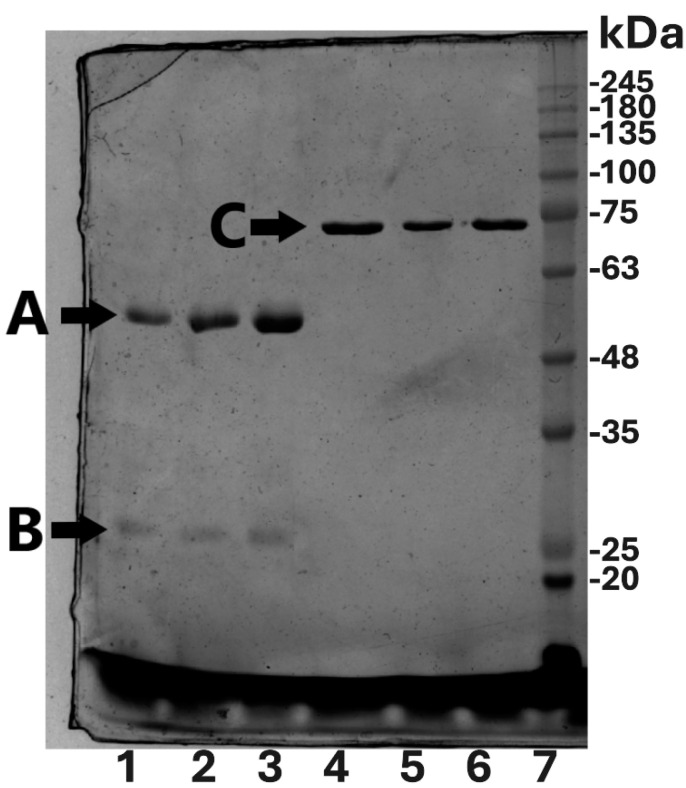
Silver-stained 10% SDS PAGE (reducing conditions) of purification products. A—IgG heavy chain; B—IgG light chain; C—Tf; lanes 1 and 4—MES final products; lanes 2 and 5—BT products; lanes 3 and 6—PB buffer products; lane 7—molecular markers.

**Figure 6 molecules-30-00993-f006:**
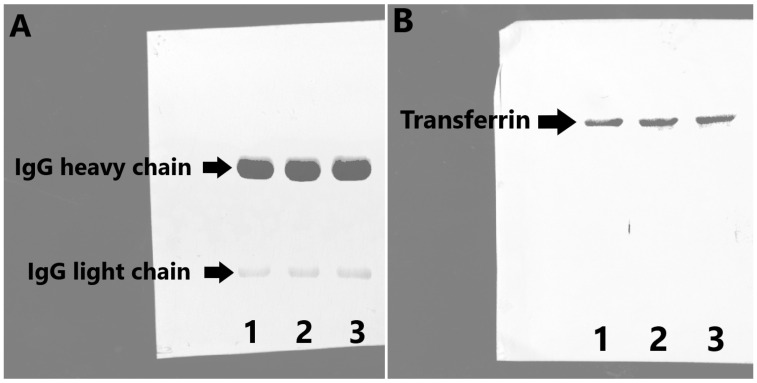
ECL visualized Western blot with immunodetection (in reducing conditions) of IgG (**A**) and of Tf (**B**). 1—BT, 2—PB, 3—MES buffer.

**Figure 7 molecules-30-00993-f007:**
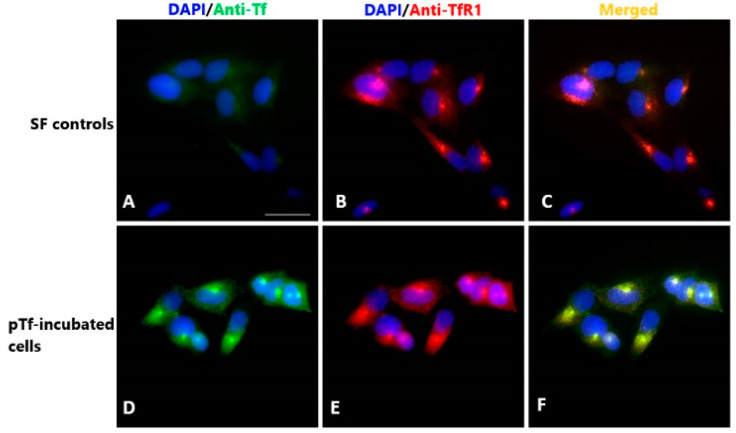
Immunolocalization of pTf and transferrin receptor 1 (TfR1) in HTR-8/SVneo cells (anti-Tf—green; anti-TfR1—red); (**A**–**C**)—SFCM controls; (**D**–**F**)—cells incubated with pTf in SFCM (pTf-incubated cells); nuclei staining—DAPI (blue); (**F**)—colocalization of pTf and TfR1 (yellow). Scale bar represents 20 µm.

**Figure 8 molecules-30-00993-f008:**
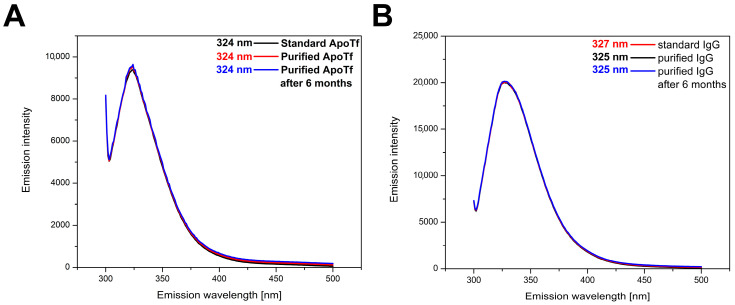
Fluorescent emission spectra of standard and purified IgG (**A**) and Tf (**B**). Also included are the fluorescence emission spectra of both proteins after 6 months of storage at −4 °C. Peak wavelengths for apoTf and IgG variants are shown in corresponding colored text. Curves were manually slightly offset for better visibility.

**Table 1 molecules-30-00993-t001:** A comparative table of the method presented in this paper and different methods for the isolation/purification of either IgG and/or Tf available in the literature.

Authors	Year	Protein	Sample	Purification Principle	Yield	Purity
Our method		IgG	Human serum	Rivanol precipitation and anion exchange chromatography	up to 28.28%	up to 99.65%
		Tf			up to 15.94%	up to 99.6%
Bresolin et al. [22]	2010	IgG	Serum	Negative chromatography (ω-aminodecyl-agarose)	≤71%	≤99%
Page et al. [23]	2002	IgG	Serum	Ion-exchange chromatography (DEAE Sepharose CL-6B)	-	>90%
Gondim et al. [24]	2012	IgG	Serum	Affinity chromatography (Cibacron F3GA immobilized onto epoxide chitosan/alginate composite)	≤53%	-
McCann et al. [20]	2005	apo-Tf	Cohn supernatant I	Ion-exchange chromatography (DEAE FF Sepharose)	55%	93%
Inman et al. [25]	1961	Tf	Cohn fraction IV-4	Precipitation techniques combined with ion-exchange chromatography	≤55%	>91%
Werner et al. [26]	1983	Tf	Serum	Affinity chromatography (DEAE Affi-Gel Blue) followed by gel chromatography or another affinity chromatography	>80%	-
Penezic et al. [27]	2017	Tf	Serum	Precipitation using rivanol, followed by two steps of ammonium sulphate precipitation	58%	97%
Ascione et al. [19]	2010	apo-Tf	Plasma	Cohn fractionation (fraction IV-4 was used) combined with ion-exchange chromatography	80%	>95%

**Table 2 molecules-30-00993-t002:** Peak area and retention time for interday and intraday reproducibility of chromatographic purification of IgG and Tf using different buffers with the CVs.

IgG	After AIEX Chromatography Using BisTris Buffer		After AIEX Chromatography Using MES Buffer		After AIEX Chromatography Using Phosphate Buffer	
	Intraday	Interday	Intraday	Interday	Intraday	Interday
Peak area, mL*mAU (CV)	75.59 (0.81)	76.20 (0.80)	94.37 (1.80)	95.13 (1.80)	122.93 (1.38)	123.92 (1.38)
Retention volume, mL (CV)	5.94 (0.97)	5.91 (1.28)	7.04 (1.41)	7.07 (0.50)	4.99 (2.69)	4.92 (1.50)
Tf	After AIEX Chromatography Using BisTris Buffer		After AIEX Chromatography Using MES Buffer		After AIEX Chromatography Using Phosphate Buffer	
	Intraday	Interday	Intraday	Interday	Intraday	Interday
Peak area, mL*mAU (CV)	45.54 (0.82)	46.49 (0.82)	27.34 (2.61)	27.91 (2.61)	22.28 (1.41)	22.75 (1.41)
Retention volume, mL (CV)	38.74 (0.05)	38.70 (0.21)	37.83 (0.11)	37.62 (0.71)	25.62 (0.61)	25.53 (0.46)

**Table 3 molecules-30-00993-t003:** Yield, purity, and purification factor of simultaneously isolated Tf and IgG by using different buffer systems.

IgG	Initial Serum Pool	After Rivanol Treatment	After AIEX Chromatography Using BisTris Buffer	After AIEX Chromatography Using MES Buffer	After AIEX Chromatography Using Phosphate Buffer
Yield ± SD (%)	100	51.54 ± 1.02	17.39 ± 0.14	21.71 ± 0.39	28.28 ± 0.39
Purity ± SD (%)	31.82 ± 1.45	30.16 ± 0.29	99.65 ± 0.07	99.45 ± 0.27	98.24 ± 0.19
Purification factor ± SD	-	0.94 ± 0.053	3.14 ± 0.05	3.13 ± 0.04	3.09 ± 0.04
Tf	Initial Serum Pool	After Rivanol Treatment	After AIEX Chromatography Using BisTris Buffer	After AIEX Chromatography Using MES Buffer	After AIEX Chromatography Using Phosphate Buffer
Yield ± SD (%)	100	23.04 ± 0.20	15.94 ± 0.13	9.57 ± 0.25	7.80 ± 0.11
Purity ± SD (%)	3.13 ± 0.06	0.72 ± 0.02	99.60 ± 0.03	99.53 ± 0.24	98.52 ± 0.10
Purification factor ± SD	-	0.23 ± 0.01	31.86 ± 0.23	31.83 ± 0.31	31.51 ± 0.49

## Data Availability

Data available upon request.

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
