# Peer review of "Simultaneous Isolation and Purification of Transferrin and Immunoglobulin G from Human Serum—A New Biotech Solution"

_molecules, 2025, doi:10.3390/molecules30050993_

Round 1
Reviewer 1 Report
Comments and Suggestions for Authors
The article aligns with the journal's scope; however, the authors should more explicitly highlight the advantages of their method compared to existing ones. To enhance clarity, it is recommended that a comparative table be created to outline the parameters of IgG and transferrin purification methods described in the literature and of presented method. The article requires significant revisions.
Critical Comment
Lines 235-236: The authors mention evaluating the method's reproducibility, but I was unable to find data confirming intraday and interday reproducibility. While standard deviations are provided in Table 1, the number of repetitions used to calculate this parameter is not specified. The authors should provide data assessing the purification method's reproducibility, including the coefficient of variation—a standard parameter used in evaluating reproducibility.
Author Response
Comment 1: The article aligns with the journal's scope; however, the authors should more explicitly highlight the advantages of their method compared to existing ones. To enhance clarity, it is recommended that a comparative table be created to outline the parameters of IgG and transferrin purification methods described in the literature and of presented method. The article requires significant revisions.
Answer 1: The authors are thankful and appreciate the time and effort invested by the Reviewer to read and comment our Manuscript. We agree with the Reviewer's comment and we have incorporated a comparative table into the revised Manuscript in order to clarify and highlight the advatages of our method compared to existing ones.
Comment 2: Lines 235-236: The authors mention evaluating the method's reproducibility, but I was unable to find data confirming intraday and interday reproducibility. While standard deviations are provided in Table 1, the number of repetitions used to calculate this parameter is not specified. The authors should provide data assessing the purification method's reproducibility, including the coefficient of variation—a standard parameter used in evaluating reproducibility.
Answer 2: We are grateful to the Reviewer for this comment. We have added the number of replicates used for the calculations, as well as, the coefficient of variation for the estimation of the method's inter/intraday reproducibility in the revised Manuscript.
Reviewer 2 Report
Comments and Suggestions for Authors
The manuscript contains a well described protocol for simultaneous purification and characterization of human transferrin and IgG. The aim of the study was to develop a time- and cost-effective isolation method. The text is easy to follow, and the description of procedures is well detailed allowing reproducibility. Here are a few details for authors' consideration:
Line 26. Typo: "hepatocites" should read "hepatocytes".
Line 75. What do the brackets stand for, perhaps should be removed?
Line 262. "Ig G" should read "IgG"
Lines 331-341. I guess that the information from acknowledgements section shall be distributed to funding and conflict of interest sections accordingly, and please take care of filling in other important points such as authors contributions and ethics approval.
Author Response
The manuscript contains a well described protocol for simultaneous purification and characterization of human transferrin and IgG. The aim of the study was to develop a time- and cost-effective isolation method. The text is easy to follow, and the description of procedures is well detailed allowing reproducibility. Here are a few details for authors' consideration:
Comment 1: Line 26. Typo: "hepatocites" should read "hepatocytes".
Answer 1: Agree. Corrected.
Comment 2: Line 75. What do the brackets stand for, perhaps should be removed?
Answer 2: Agree. Corrected.
Comment 3: Line 262. "Ig G" should read "IgG"
Answer 3: Agree. Corrected.
Comment 4: Lines 331-341. I guess that the information from acknowledgements section shall be distributed to funding and conflict of interest sections accordingly, and please take care of filling in other important points such as authors contributions and ethics approval.
Answer 4: Agree. We have added the missing information, though these have already been disclosed to the Publisher during the submission process. Authors contributions were also added during the submission.
The authors thank the Reviewer for the time and effort put in this review.
Reviewer 3 Report
Comments and Suggestions for Authors
I have reviewed the article titled “Simultaneous isolation and purification of transferrin and immunoglobulin G from human serum – a new biotech solution” by Citec et al. The simultaneous isolation and purification of transferrin (Tf) and immunoglobulin G (IgG) from the same serum sample is indeed a novel approach. However, the manuscript could benefit from greater emphasis on its practical applications, such as its potential for theranostic use in clinical and research settings. I have some comments that may help improve the article.
The study confirms that both proteins retain their native structure and functionality post-purification. While this finding is important, additional experiments testing the reproducibility of these results across larger sample sets would strengthen the manuscript.
Major Comments
The use of ion-exchange chromatography with flexibility in buffer selection is commendable. However, the relatively low yields of Tf and IgG raise concerns about the efficiency of the precipitation and purification processes. This issue should be explicitly addressed in the discussion. Have the authors considered investigating alternative precipitation agents or buffer compositions to increase the yields of Tf and IgG while minimizing protein loss?
The SDS-PAGE and chromatographic data are clear, but the manuscript lacks detailed statistical analysis of reproducibility and variation between replicates. Including standard error margins or variability metrics, such as an ANOVA test, for each buffer system would enhance the robustness of the study.
Although the MS briefly mentions low yields as a limitation due to rivanol precipitation, a more thorough discussion of other potential limitations—such as protein loss during subsequent steps or the scalability of the method for industrial applications—would improve the paper.
Based on the authors' statement that "the proposed method enables the production of two theranostic-grade proteins applicable in research, analytical procedures, diagnosis, and therapy from the same sample simultaneously…" I suggest repeating the purification process using serum from diverse populations or patients with specific conditions (e.g., cancer or autoimmune diseases). This would test the generalizability and applicability of the method across different diseases.
The MS does not address the long-term stability of the purified Tf and IgG. Experiments assessing whether the purified proteins maintain their native structures and functionalities over extended storage periods would provide valuable insights.
The authorss could explore potential downstream applications of purified Tf and IgG in greater depth. For instance, experiments to test Tf's ability to deliver iron in cell culture models or therapeutic scenarios would be valuable. Similarly, additional immunogenicity assays, such as cytokine release tests, could confirm IgG's functionality in immune activation.
Testing the isolated proteins in diagnostic assays, such as ELISA or flow cytometry, would validate their compatibility and effectiveness in commonly used clinical tests.
Confocal microscopy could provide better visualization for Figure 5. The authors may use the same slides prepared for this experiment to obtain improved images. Additionally, Figure 3 and Figure 4 should be revised for publication quality, and complete gels and blots should be provided in the supplementary materialss.
A flowchart summarizing the workflow and key findings of the study would improve the visual communication of the methodology and results.
minor comments
Improve figure captions by including more detailed descriptions of the data and results presented.
Please rename the “Results” section to “Results and Discussion” to better reflect the content.
Author Response
Comment 1: I have reviewed the article titled Simultaneous isolation and purification of transferrin and immunoglobulin G from human serum - a new biotech solution by Citec et al. The simultaneous isolation and purification of transferrin (Tf) and immunoglobulin G (IgG) from the same serum sample is indeed a novel approach. However, the manuscript could benefit from greater emphasis on its practical applications, such as its potential for theranostic use in clinical and research settings. I have some comments that may help improve the article.
Answer 1: We are thankful to the Reviewer for the time and effort to read and review our paper.
Comment 2: The study confirms that both proteins retain their native structure and functionality post-purification. While this finding is important, additional experiments testing the reproducibility of these results across larger sample sets would strengthen the manuscript.
Answer 2: We agree with the Reviewer that a larger set of samples would strengthen the manuscript, so we have incorporated the results obtained upon the application of our method for the isolation and purification of IgG and Tf from diabetic population. We have also applied this method in the case of peritoneal dialysis patients, and achieved the same results, but at this moment we are not at liberty to share these results.
Major Comments
Comment 3: The use of ion-exchange chromatography with flexibility in buffer selection is commendable. However, the relatively low yields of Tf and IgG raise concerns about the efficiency of the precipitation and purification processes. This issue should be explicitly addressed in the discussion. Have the authors considered investigating alternative precipitation agents or buffer compositions to increase the yields of Tf and IgG while minimizing protein loss?
Answer 3: We are thankful for this comment. We are considering other protein precipitation agents instead of rivanol, as well as, other methods for rivanol removal in order to escape the use of activated charcoal, which is the main reason for relatively low yield of both isolated proteins. Our idea from the beginning was to ensure extremely high purity of the isolates, but now we are undertaking further experiments to improve the yields. We have added this information in the revised Manuscript.
Comment 4: The SDS-PAGE and chromatographic data are clear, but the manuscript lacks detailed statistical analysis of reproducibility and variation between replicates. Including standard error margins or variability metrics, such as an ANOVA test, for each buffer system would enhance the robustness of the study.
Answer 4: We are very grateful to the Reviewer for pointing this out, we have added the missing statistical parameters in the revised Manuscript.
Comment 5: Although the MS briefly mentions low yields as a limitation due to rivanol precipitation, a more thorough discussion of other potential limitations such as protein loss during subsequent steps or the scalability of the method for industrial applications would improve the paper.
Answer 5: Agree. The authors are grateful for this comment. We have expanded the discussion regarding other limitations i.e. the proteins loss due to the application of activated charcoal for the rivanol removal, and during the chromatographic step of our method.
Comment 6: Based on the authors' statement that "the proposed method enables the production of two theranostic-grade proteins applicable in research, analytical procedures, diagnosis, and therapy from the same sample simultaneously;" I suggest repeating the purification process using serum from diverse populations or patients with specific conditions (e.g., cancer or autoimmune diseases). This would test the generalizability and applicability of the method across different diseases.
Answer 6: The authors wish to thank for this comment. We have added the results obtained upon application of our method in order to isolate and purify Tf and IgG from diabetic population. As mentioned before, we have conducted the same procedure in the case of patients on peritoneal dialysis, but at the moment we are not in liberty to publish them.
Comment 7: The MS does not address the long-term stability of the purified Tf and IgG. Experiments assessing whether the purified proteins maintain their native structures and functionalities over extended storage periods would provide valuable insights.
Answer 7: We are thankful for this comment. We have added the data about the stability upon storage for 6 months at +4 °C in the revised Manuscript.
Comment 8: The authors could explore potential downstream applications of purified Tf and IgG in greater depth. For instance, experiments to test Tf's ability to deliver iron in cell culture models or therapeutic scenarios would be valuable. Similarly, additional immunogenicity assays, such as cytokine release tests, could confirm IgG's functionality in immune activation.
Answer 8: The authors are grateful for this comment, yet at this point we are interested in the production of extremely pure proteins in a fast and affordable fashion and in sufficient quantities, since both of the proteins have a great theranostic value. The experiments for the iron delivery are already underway (Tf as a therapeutic and a mean for targeted delivery) but they are a part of a different study. Regarding the cytokine release assay for the IgG, the authors unfortunately lack the means to conduct such an experiment, but the immunogenicity has been confirmed in-house – we have successfully immunized sheep and had a very good titer of anti-IgG antibodies. These results are not presented in the paper since they are a part of INEPs commercial activities and fall into protected IP category. If it is to the Reviewers wishes, the authors are more than willing to send these data upon an official request.
Comment 9: Testing the isolated proteins in diagnostic assays, such as ELISA or flow cytometry, would validate their compatibility and effectiveness in commonly used clinical tests.
Answer 9: Agree. Unfortunately, setting up an ELISA or flow cytometry test for the validation of our isolates requires certain (not short) amount of time as well as gathering a significant number of samples in order to have an appropriate power of the study. We have designed such a study already, but we are in the process of acquiring the samples, tuning of the parameters and testing of protocols. We can disclose with the Reviewer that the IgG isolated by our method was used for testing of patients’ samples by the Blood Transfusion Institute of Serbia (again, the authors are not in liberty to share results regarding commercial activities of the INEP Institute).
Comment 10: Confocal microscopy could provide better visualization for Figure 5. The authors may use the same slides prepared for this experiment to obtain improved images. Additionally, Figure 3 and Figure 4 should be revised for publication quality, and complete gels and blots should be provided in the supplementary materials.
Answer 10: The authors couldn’t agree more. Unfortunately, the Institute does not posses a confocal microscope and also unfortunately the waiting list for the one that we can access is far too long for the amount of time we were given to finish this review. Figures 3 and 4 were revised for the publication quality, and complete gels and blots are going to be provided to the Publisher as supplementary material. Again, if the Reviewer wishes, we can provide these upon request even before the Publisher takes action.
Comment 11: A flowchart summarizing the workflow and key findings of the study would improve the visual communication of the methodology and results.
Answer 11: Agree. The authors have designed and added a flowchart summarizing the workflow and key findings in the revised Manuscript.
minor comments
Comment 12: Improve figure captions by including more detailed descriptions of the data and results presented.
Answer 12: We are grateful for this comment, we have improved the figure captions accordingly.
Comment 13: Please rename the Results section to Results and Discussion to better reflect the content.
Answer 13: Agree. We have made the suggested change in the revised Manuscript.
Round 2
Reviewer 1 Report
Comments and Suggestions for Authors
The revision is Ok
Reviewer 3 Report
Comments and Suggestions for Authors
The authors have adequately addressed my concerns and have explained the limitations of their study. I consider it suitable for publication.